# Three Questions Concerning the Use of Large Language Models to Facilitate Mathematics Learning

**An-Zi Yen**[*] and **Wei-Ling Hsu**

Department of Computer Science, National Yang Ming Chiao Tung University, Taiwan
azyen@nycu.edu.tw, weiling.hsu.cs11@nycu.edu.tw

## Abstract

Due to the remarkable language understanding and generation abilities of large language models (LLMs), their use in educational applications has been explored. However, little work has been done on investigating the pedagogical ability of LLMs in helping students to learn mathematics. In this position paper, we discuss the challenges associated with employing LLMs to enhance students' mathematical problem-solving skills by providing adaptive feedback. Apart from generating the wrong reasoning processes, LLMs can misinterpret the meaning of the question, and also exhibit difficulty in understanding the given questions' rationales when attempting to correct students' answers. Three research questions are formulated.

## 1 Introduction

After the pandemic, e-learning has become part of mainstream education (Alqahtani and Rajkhan, 2020; Jafar et al., 2022). However, for students, online learning is not without its problems (Abdur Rehman et al., 2021). Apart from the difficulty in maintaining focus in online classes, the lack of real-time communication and timely feedback are also serious problems. In particular, in online assessments, students may fail to understand their mistakes, even after reviewing the provided answers; such failure to immediately clarify points of confusion yields poor learning outcomes. Synchronous communication between teachers and students in an e-learning setting is necessary, but teachers find that promptly responding to students' questions is a significant challenge.

As large language models (LLMs) offer a wide range of applications, several studies (Tack and Piech, 2022; Kasneci et al., 2023; Zhang et al., 2023) address the use of LLMs in education. In this work, we seek to investigate the integration of LLMs into education. Since each subject has its own issues that must be addressed, we explore the scenario of LLM use in mathematics education and the challenges thereof. Many studies address math word problem solving (Shen et al., 2021; Yu et al., 2021; Jie et al., 2022), but the human utility of the mathematics reasoning processes and the natural language explanations generated by models within educational settings is rarely discussed. Given the extraordinary capability of LLMs to generate free-text rationalization, we investigate their mathematical problem-solving competence and assess whether the generated step-by-step explanations constitute a useful educational resource. In particular, we analyze the ability of LLMs to provide adaptive feedback by identifying and explaining errors within a student's math problem-solving process.

We address these issues using the MathQA dataset (Amini et al., 2019), which consists of GRE-level questions that require advanced mathematical knowledge. The questions cover arithmetic, algebra, geometry, and data analysis. In Appendix A we explain in detail our reasoning behind the dataset selection. For all experiments, we use GPT-3.5 (Ouyang et al., 2022). We raise and discuss a total of three research questions. In this pilot study, we conduct both quantitative and qualitative evaluations to answer the research questions. The contributions of this work are threefold.

1. We explore the application of LLMs for instructing students in solving math word problems.
2. Drawing on the results of LLMs to solve math word problems, we comprehensively identify the existing problems of current models.
3. We discuss the pedagogical suitability of the generated equations and the corresponding free-text explanations in the context of mathematics education.

---

[*]Corresponding author.

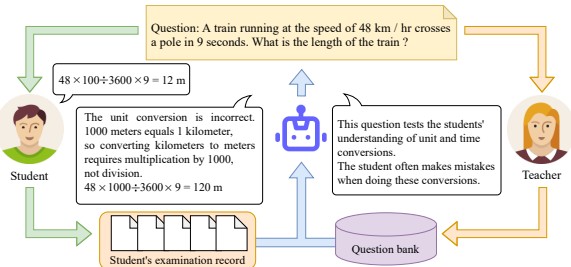

Figure 1: An LLM helping a student to learn mathematics

| Category | Questions | Zero-shot | Few-shot | CoT |
|---|---|---|---|---|
| All | 1,605 | **66.54%** | 65.67% | 66.11% |
| General | 663 | 64.71% | 63.20% | **64.86%** |
| Gain | 345 | 71.88% | **72.17%** | 70.14% |
| Physics | 410 | **68.54%** | 65.37% | 67.32% |
| Geometry | 100 | 63.00% | **65.00%** | 60.00% |
| Probability | 12 | 58.33% | **83.33%** | 75.00% |
| Other | 75 | 53.33% | 57.33% | **58.67%** |

Table 1: MathQA results

## 2 Utility of LLMs in Mathematics Learning

To explore the utility of LLMs in mathematics learning, we raise the first research question (**RQ1**): How can LLMs be utilized to assist students in learning mathematical problem-solving skills? Since exams are a common way for teachers to evaluate student learning progress, in this study, we focus on leveraging LLMs to equip students with the knowledge skills needed to solve math word problems.

In terms of question types, in addition to true/false and multiple-choice questions, short-answer questions are also included in math exams. Students answer the questions using problem-solving processes. Automated assessment (Moore et al., 2022) of the student's problem-solving process remains to be investigated. In education, feedback is also crucial (Shute, 2008). Simply scoring the student answer is often insufficient, as scores do not reflect the reasons for incorrect answers. To learn from their mistakes, students need the corresponding explanation. Figure 1 illustrates the scenario of an LLM applied in mathematics education. After a student responds to a question, the model determines whether the student's answer is correct or not. If the answer is correct, the system informs the student of the test objective for that question, enhancing their understanding of the examined skills. If the answer is incorrect, the system provides adaptive feedback (Bernius et al., 2022; Sailer et al., 2023) by indicating the location of the error and offering an explanation, assisting the student to clarify any misunderstanding.

Given that there are multiple methods for solving a math word problem, a model must be able to understand and correct a student's thought process. If the overall strategy is sound apart from minor concept errors or obstacles, the model should guide students to the next step following their chosen approach. Thus the ability of LLMs to solve math word problems and to understand and explain equations is critical. The following sections address these two key points.

## 3 LLM Ability to Solve Math Problems

Many textbook math exercises provide the calculation process but lack a natural language explanation. Accompanying each step with an explanation would greatly enhance the student's understanding of each equation. This leads to the second research question (**RQ2**): What is the problem-solving capability of LLMs in mathematical problems and their ability to explain the computational process? **Automatic Evaluation:** The dataset used in this work is a modified version of MathQA in which unsolvable questions were removed by Jie et al. (2022). We utilized OpenAI's API, in particular the "gpt-3.5-turbo-0301" model. The temperature was set to 0. Table 1 shows the accuracy of three commonly used prompting methods: zero-shot prompting (Zero-shot), few-shot prompting (Few-shot), and chain-of-thought (CoT) prompting. Three prompting templates are shown in Appendix B. "Questions" denotes the number of questions of each type in the test set. We also show the results for six MathQA question types. In this experiment, we compared only the model-generated answers with the provided answers, without verifying the correctness of the reasoning process. As shown in Table 1, the CoT performance was poorer than expected, possibly due to the higher mathematical skill demands of MathQA compared to previously-used datasets. Zero-shot prompting does not significantly outperform CoT ($p < 0.8$). Exploring suitable prompting methods is left as future work. Based on the results, we observe that the calculation abilities of the GPT-3.5 model remain far from satisfactory. It frequently fails at simple arithmetic operations or counting. **Human Evaluation of LLM Results:** It is known that LLMs might produce the correct answer even

| Error type | Percentage |
|---|---|
| Misconception in problem-solving | 36.54% |
| Incorrect provided answer* | 17.31% |
| Unclear question definition* | 11.54% |
| Calculation error in equation | 9.61% |
| Misinterpretation of question | 7.69% |
| Arithmetic error | 5.77% |
| Absence of necessary diagrams* | 3.85% |
| Counting error | 3.85% |
| Undefined symbols in question* | 1.92% |
| Incomplete problem-solving | 1.92% |

Table 2: Error types annotated by an expert. * indicates that the error is not from the model's response but from an error in the question.

with incorrect reasoning, or give an incorrect answer despite correct reasoning (Laskar et al., 2023; Lightman et al., 2023). However, detailed analyses and statistical evaluations of model errors have been less extensively studied. To further analyze whether LLMs are able to reason through complex mathematical problems, we invited an expert who majored in Mathematics to evaluate the answers generated by the GPT-3.5 model. A total of 120 questions—20 from each of the six question types—were selected by the expert. Each selected question involved a process of reasoning and calculation of more than four steps. The error types are shown in Table 2. The most common error made by the model was "misconception in problem-solving": the model understood the question but used incorrect formulas or methods to solve it. "Misinterpretation of the question", in turn, is a different error: the model does not understand the question and generates an unrelated result. We also find that the GPT-3.5 model is good at providing comprehensive explanations for each equation without omitting parts of the problem-solving process. However, it exhibits inconsistent calculations, often making simple arithmetic errors. Furthermore, its grasp of set theory and three-dimensional spatial reasoning is limited. Examples of some error types are provided in Appendix C.

**Research Issues of Augmented Language Models:** Using external tools for calculation may be one way to address LLM drawbacks. Mialon et al. (2023) refer to language models that utilize external tools (Gao et al., 2022; Liu et al., 2022), retrieve relevant information (Izacard et al., 2022), or use specific reasoning strategies (Wei et al., 2022) as augmented language models (ALMs). We argue that for a model to solve more complex tasks, it should comprehend the tasks and know when, how,

and why to request augmentation. Otherwise, the improvements yielded by the augmented information would remain limited in various real-world applications. For instance, for mathematical calculations, Schick et al. (2023) propose having the LLM use a calculator API to solve arithmetic tasks. However, the propensity of LLMs to misinterpret question meanings and substitute incorrect numbers remains a prominent challenge in advanced mathematical reasoning. From our observation, the GPT-3.5 model behaves much like a student focused on formula memorization in that it struggles to adapt to variations in the question, particularly in probability questions. Hence, enhancing the mathematical reasoning capabilities of LLMs is a critical research direction.

## 4 Pedagogical Ability of LLMs to Rectify Students' Answers

The most important thing when helping students to learn mathematical problem-solving is providing immediate adaptive feedback. In this section, we measure the pedagogical ability of LLMs in mathematics. Pedagogical ability refers to the ability to understand and help the student (Tack and Piech, 2022). This leads to the third research question (**RQ3**): Are LLMs able to identify errors in students' answers and provide corresponding explanations?

**Teacher–Student Framework for Quantitative Evaluation:** Due to the difficulty in obtaining real-world student answers, we simulate a scenario in which students answer questions and teachers correct the students' responses. Based on the experimental results from Table 1, we use the responses from zero-shot prompting as the student answers. These answers are then input into the GPT-3.5 model, which acts as a teacher, correcting the student's answers according to the question. The GPT-3.5 model is tasked with identifying whether the student's answer is correct and explaining why. Specifically, given an input question $q$, prompt $\mathcal{P}_s$, and model $\mathcal{M}$, we obtain the initial problem-solving result $y_s = \mathcal{M}(q; \mathcal{P}_s)$. Next, we input $y_s$ to $\mathcal{M}$, and ask $\mathcal{M}$ to act as a teacher with prompt $\mathcal{P}_t$ to correct $y_s$ based on $q$. Finally, we obtain the feedback $y_t = \mathcal{M}(q, y_s, r; \mathcal{P}_t)$, where $r$ is the rationale of $q$ provided in MathQA. If $\mathcal{M}$ struggles to understand its responses (with detailed processes and natural language explanations), then its potential to assist teachers in corrections be-

| Type | W/o rationale | W/ rationale |
|---|---|---|
| All | 53.96% | 73.71% |
| General | 54.75% | 73.30% |
| Gain | 54.78% | 73.33% |
| Physics | 50.00% | 72.20% |
| Geometry | 60.00% | 83.00% |
| Probability | 25.00% | 83.33% |
| Other | 61.33% | 73.33% |

Table 3: Results of teacher–student framework

| Result of $y_s$ | $y_s$ is correct | | $y_s$ is incorrect | |
|---|---|---|---|---|
| $\mathcal{P}_t$ | w/o $r$ | w/ $r$ | w/o $r$ | w/ $r$ |
| Identify $y_s$ is correct | 63.24% | 10.29% | 53.85% | 0.00% |
| Say in other words | 17.65% | 67.65% | 17.31% | 17.31% |
| Correct the process | 11.76% | 16.18% | 11.53% | 50.00% |
| Correct the calculation | 7.35% | 5.88% | 17.31% | 32.69% |

Table 4: Error types in teacher–student framework

comes questionable. The $\mathcal{P}_t$ template is shown in Appendix B. We refer to this framework as the "teacher–student framework".

Table 3 presents the results of the teacher–student framework. Accuracy is adopted as the evaluation metric: accuracy measures whether $\mathcal{M}$ correctly identifies the correctness of $y_s$. As $y_t$ could be a lengthy paragraph, we simply use keywords such as "is correct", "is incorrect", "not correct", "almost correct" or "not correct" to identify $\mathcal{M}$. We compare the results with and without the question rationales. Interestingly, if the corresponding rationales are not given as input, the accuracy of the GPT-3.5 model acting as a teacher to correct students' answers (53.96%) is lower than that when directly answering questions (66.54%). However, if the corresponding rationales are given as input, accuracy is only 73.71%. Thus the GPT-3.5 model has difficulty understanding the equations given in the rationales.

**Human Evaluation of Correcting Model-Generated Answers:** To understand why the GPT-3.5 model exhibits these characteristics when correcting answers, our expert also analyzed the results of the teacher–student framework based on the selected 120 questions. Examples are given in Appendix D. Table 4 presents the correction results $y_t$ for $y_s$ by human evaluation. "w/o $r$" and "w/ $r$" denote $\mathcal{P}_t$ without and with rationales, respectively. Comparing the results of "w/o $r$" and "w/ $r$", we find that providing rationales seriously confuses $\mathcal{M}$, such that it determines $y_s$ is wrong in most cases. Furthermore, $\mathcal{M}$ simply rephrases the content of $y_s$ (67.65%). Note that the results in Table 3

show that $\mathcal{P}_t$ with $r$ is better than that without $r$. However, the results in Table 4 are different, because the questions selected by the expert are more challenging. If $y_s$ is incorrect, $\mathcal{M}$ has a 53.85% chance of erroneously claiming that $y_s$ is correct when $r$ is not provided. When $r$ is given, $\mathcal{M}$ correctly identifies that $y_s$ is incorrect. Nonetheless, $\mathcal{M}$ has only a 10.29% chance to accurately identify $y_s$ as correct with $r$. In addition, according to our statistic, $\mathcal{M}$ has only a 3.85% and 1.92% chance to accurately correct the calculation results and the problem-solving processes, respectively. This is primarily because it has difficulty exploiting $r$ and because it usually misunderstands the $y_s$ equations, even to the point of inaccurately correcting those that are already correct. Furthermore, it often forgets the equations and calculations in $y_s$. To verify whether LLMs can understand and correct human problem-solving processes, we also invited five college students to answer three questions. The results are presented in Appendix E.

**Utility of LLMs in Complex Reasoning Tasks for Education:** The failures of the GPT-3.5 model are comprehensively analyzed by Borji (2023). In this work, we find that LLMs tend to be confused by human answers, especially when tasks demand advanced knowledge or reasoning skills, even if relevant and correct information is provided. Hence, an alternative framework is needed under which to leverage LLMs' language understanding ability in complex reasoning tasks. Other crucial research issues are how to make the models aware of what they do not know and how to produce truthful and interpretable results (Phillips et al., 2020). Besides, this work primarily focuses on the capabilities and challenges of directly using LLMs for correcting students' answers. Developing a cognitive model for reasoning processes and their potential roles in rectifying student mistakes is crucial.

## 5 Conclusion

In this work we propose a research agenda for leveraging LLMs in mathematics learning. First, we explore the use of LLMs to assist students in learning math word problem-solving skills. Then, we analyze the mathematical reasoning ability of LLMs. Finally, we investigate the pedagogical ability of LLMs in terms of rectifying model-generated or human answers and offering adaptive feedback. We conduct experiments with the GPT-3.5 model, and conclude that there remains room for improve-

ment in the LLM's performance in solving complex mathematical problems. In addition, although it generates comprehensive explanations, the LLM is limited in accurately identifying model's and human's errors due to its poor ability to interpret mathematical equations. In the future, we plan to develop an advanced method by which to improve the utility of LLMs in mathematics education.

## Limitations

Considering that LLMs are widely accessible to the general public and many educational institutions are examining the challenges and benefits of their use by teachers and students, this paper primarily focuses on the application of LLMs in educational settings. However, our experiments employed only the GPT-3.5 model and did not explore other LLMs such as GPT-4 (OpenAI, 2023) and LLaMA (Touvron et al., 2023). Furthermore, while our current work investigates the utility of LLMs in enhancing students' mathematical problem-solving skills, there are many other applications in education. For instance, LLMs could be utilized to help teachers to generate teaching content or questions. Additionally, the potential issues of violating teaching ethics involved in introducing LLMs into educational applications are significant topics that have not been included in this study currently. We also conducted a human evaluation on 120 questions, which may be insufficient. Although we invited the expert to select questions that cover as many types of questions as possible, there may remain worthwhile examples that were not selected for analysis. Moreover, the mistakes made by humans and those made by LLMs may differ. However, at the current stage, we invited only five college students to answer three questions: a larger-scale experiment would be more helpful.

## Ethics Statement

In the context of educational applications, we utilize students' personal data and answer records for our experiments, which raises privacy concerns. For this study, we invited college students to answer mathematical questions to investigate issues that might be found in real-world applications. These participants fully understood how their data would be used, and we ensured that their personal information would not be leaked.

## Acknowledgements

This research was partially supported by National Science and Technology Council, Taiwan, under grant NSTC 111-2222-E-A49-010-MY2.

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

## A   Dataset Selection

Datasets commonly used to evaluate a model's ability to solve math word problems include MathQA (Amini et al., 2019), SVAMP (Patel et al., 2021), Math23K (Wang et al., 2017), and GSM8K (Cobbe et al., 2021). GSM8K consists of grade-school math word problems, and SVAMP primarily covers up to grade 4. Math23K is designed for Chinese elementary school students, and focuses on evaluating the ability of students to solve one-unknown-variable linear math word problems. By contrast, MathQA is composed of GRE-level math problems, which serve as a benchmark for university research institute admissions in the United States. Therefore, we use MathQA in our experiments to examine the utility of LLMs in higher education applications.

| Input: You are a student who is good at mathematics. Apply the appropriate mathematical operations to solve the math problem. Please choose an answer from the options. |
| --- |
| Q: a student gets 55 % in one subject , 65 % in the other . to get an overall of 55 % how much should get in third subject . Options: a ) 75 % , b ) 25 % , c ) 45 % , d ) 55 % , e ) 65 % |
| **Output:** Let's assume that the third subject carries the same weight as the other two subjects. To get an overall of 55%, the average of the three subjects should be 55%. Let x be the percentage the student needs to get in the third subject. Then, we can set up the equation: $(55\% + 65\% + x)/3 = 55\%$ Simplifying this equation, we get: $120\% + x = 165\%$ $x = 45\%$ Therefore, the student needs to get 45% in the third subject to get an overall of 55%. The answer is c) 45%. |

Table 5: Zero-shot prompting

## B   Input Formats

LLMs are known for their success in zero-shot or few-shot learning (Brown et al., 2020). We can solve various tasks (Liu et al., 2023) through prompt engineering by providing a few examples or instructions about the task descriptions. For solving math word problem, Wei et al. (2022) propose chain-of-thought prompting (CoT) by providing step-by-step reasoning examples to cause the model to decompose the given question and generate reasoning steps. They demonstrate that CoT achieves promising results in solving math word problems. Moreover, Wu et al. (2023) demonstrate the CoT's effectiveness by incorporating it into the medical diagnostic framework. Hence, we use zero-shot, few-shot, and CoT prompting in our experiments. Tables 5, 6, and 7 contain templates of zero-shot, few-shot, and CoT prompting, respectively. The template of the proposed teacher–student framework is shown in Table 8.

## C   Examples Error Types when Solving Math Word Problems

Among the 120 selected questions, the model correctly answers 68 questions and incorrectly answers 52 questions. Eighteen of the incorrect responses are due to unclear question definitions, 24 stem from errors in the model's problem-solving process, and 10 are caused by calculation errors. In this section, we will present errors made by the GPT-3.5 model to show its limitations in mathematics reasoning.

| **Input:** You are a student who is good at mathematics. Apply the appropriate mathematical operations to solve the math problem. Please choose an answer from the options. 
 Q: the banker ' s gain of a certain sum due 3 years hence at 10 % per annum is rs . 36 . what is the present worth ? Options: a ) 38 , b ) 27.675 , c ) 30 , d ) data inadequate , e ) none of these 
 A: The answer is a. 
 Q: sophia finished 2 / 3 of a book . she calculated that she finished 90 more pages than she has yet to read . how long is her book ? Options: a ) 229 , b ) 270 , c ) 877 , d ) 266 , e ) 281 
 A: The answer is b. 
 Q: a student gets 55 % in one subject , 65 % in the other . to get an overall of 55 % how much should get in third subject . Options: a ) 75 % , b ) 25 % , c ) 45 % , d ) 55 % , e ) 65 % |
|---|
| **Output:** Let's assume that all three subjects carry equal marks. 
 To get an overall percentage of 55%, the sum of the percentages obtained in all three subjects should be 3 x 55% = 165%. The student has already obtained 55% and 65% in the first two subjects. 
 Therefore, the percentage required in the third subject to get an overall percentage of 55% is: 
 Percentage required in the third subject = 165% - (55% + 65%) = 45% 
 Hence, the answer is option c) 45%. |

Table 6: Few-shot prompting

| **Input:** You are a student who is good at mathematics. Apply the appropriate mathematical operations to solve the math problem. Please choose an answer from the options. 
 Q: a student gets 55 % in one subject , 65 % in the other . to get an overall of 55 % how much should get in third subject . Options: a ) 75 % , b ) 25 % , c ) 45 % , d ) 55 % , e ) 65 % 
 A: Let's think step by step. |
|---|
| **Output:** To get an overall of 55%, the average of the three subjects should be 55%. 
 Let's assume the third subject has a weightage of 100%. 
 So, the total weightage of the three subjects is 100% + 100% + 100% = 300%. 
 Now, we know that the student got 55% in the first subject and 65% in the second subject. 
 So, the total marks obtained in the first two subjects are: 55% + 65% = 120% 
 To get an overall of 55%, the total marks obtained in all three subjects should be: 
 55% x 3 = 165% 
 Now, we can find out how much the student should get in the third subject: 
 Total marks in all three subjects = Marks obtained in the first two subjects + Marks obtained in the third subject 
 165% = 120% + Marks obtained in the third subject 
 Marks obtained in the third subject = 165% - 120% = 45% 
 Therefore, the answer is option c) 45%. |

Table 7: CoT prompting

As shown in Table 2, the most common mistake is misconception in problem-solving. Moreover, as mentioned in Section 3, one weakness of the GPT-3.5 model is in reasoning about three-dimensional spatial problems, as illustrated in Figure 3. The model directly divides the length, width, and height by three to calculate how many cubes can be placed in a box, without considering the fact that cubes cannot be accommodated if the height is not evenly divisible. The question in the red box should be $3 \times 3 \times 2$, not $4 \times 3 \times 2$.

Misinterpretation of the question is another critical error. Figure 4 is an example of the GPT-3.5 model misinterpreting the meaning of the question. Based on the result responded by the model, once we obtain the value of "$x$", the greater number is "$2x$". However, the model incorrectly interprets the question as asking which number is greater after adding five.

## D  Example Results from Teacher–Student Framework

According to our observations, when acting as a teacher, the GPT-3.5 model achieves high accuracy in correcting simple equations when the rationale contains a single equation. However, for more complex problems involving two or more equations, the quality of the rationale significantly impacts the

model's identification results. Figure 2 shows the confusion matrix of answer correction. We define the difficulty of the questions based on the number of equations in the given rationales. We categorize the question difficulty into three groups: those with fewer than five equations, those with 5 to 10 equations, and those with more than 11 equations. "Correct" and "wrong" in the true labels indicate that $y_s$ is correct and incorrect, respectively. "Correct" and "wrong" in the predicted labels label indicate that the identification of $y_t$ is correct and incorrect, respectively. Thus "correct" in the predicted labels indicates that the model identifies $y_s$ to be correct when $y_s$ is indeed correct. As shown in Figure 2, the model's accuracy in correcting answers decreases as the problem becomes more difficult, especially when no rationale is provided to the model. Clear and explicit rationales aligned with the problem-solving process in $y_s$ cause the teacher model to understand correctly and thus accurately identify whether the $y_s$ is correct.

Additionally, when $y_s$ is correct, the GPT-3.5 model frequently identifies $y_s$ as incorrect but its explanation is merely a paraphrase of $y_s$. The reason may be that the GPT-3.5 model does not merely determine correctness based on the answer's value. Although it has stringent requirements for this process, its comprehension of equations is less than ideal, resulting in misinterpretations.

**Input:** You are a math teacher. According to **[Question]** and **[Rationale]**, please indicate whether the **[Student's Answer]** is correct or not, and apply appropriate mathematical operations to correct the **[Student's Answer]**.
**[Question]**: a student gets 55 % in one subject , 65 % in the other . to get an overall of 55 % how much should get in third subject . Options: a ) 75 % , b ) 25 % , c ) 45 % , d ) 55 % , e ) 65 %
**[Rationale]**: explanation : t = 3 years r = 10 % td = ( bg × 100 ) / tr = ( 36 × 100 ) / ( 3 × 10 ) = 12 × 10 = rs . 120 td = ( pw × tr ) / 100 => 120 = ( pw × 3 × 10 ) / 100 => 1200 = pw × 3 pw = 1200 / 3 = rs . 400 answer : option a
**[Student's Answer]**: Let's assume that the third subject carries the same weight as the other two subjects.
To get an overall of 55%, the average of the three subjects should be 55%.
Let x be the percentage the student needs to get in the third subject.
Then, we can set up the equation:
(55% + 65% + x)/3 = 55%
Simplifying this equation, we get:
120% + x = 165%
x = 45%
Therefore, the student needs to get 45% in the third subject to get an overall of 55%.
The answer is c) 45%.

**Output:** The answer is correct.
Explanation:
To find out how much the student should get in the third subject to have an overall of 55%, we can use the formula:
(55% + 65% + x)/3 = 55%
We can simplify this equation by multiplying both sides by 3:
55% + 65% + x = 165%
Now we can solve for x by subtracting 55% and 65% from both sides:
x = 165% - 55% - 65%
x = 45%
Therefore, the student needs to get 45% in the third subject to have an overall of 55%.

Table 8: Prompting under the teacher–student framework

To measure the ability of LLMs to correct students' answers, we utilize the problem-solving results returned by the GPT-3.5 model as the students' answers and then ask the model to correct the input answers. Figure 5 presents an example in which the reasoning in the answer is not finished: it stops at calculating the total volume of the drinks. The model, which acts as a teacher, accurately points out the error and provides the complete solution. However, the GPT-3.5 model may also mistakenly rectify correct answers as incorrect. Taking Figure 6 as an example, the result in the given answer correctly solves for the number of sheep, yet the model identifies it as incorrect, further producing an erroneous reasoning process.

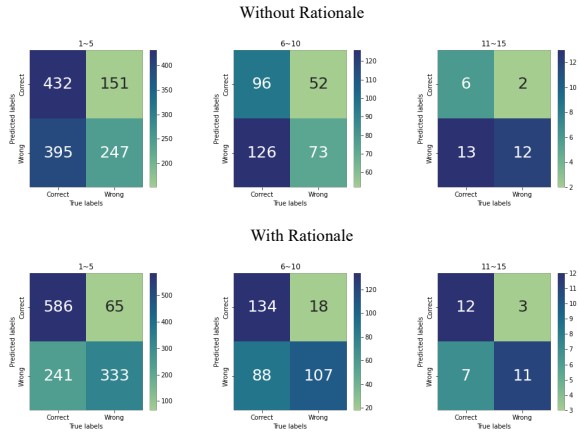

Figure 2: Confusion matrix of answer correction based on question difficulty

## E Human Evaluation on Correcting Human's Answers

Since the problem-solving approaches of LLMs may differ from those of humans, to verify whether LLMs can understand and correct human problem-solving processes, we also invited five college students from different departments (Mathematics, Statistics, and Japanese) to answer three questions selected by the expert. Their answers were input into the GPT-3.5 model for correction. Subsequently, the expert analyzed the correction results. Based on the collected answers, we find that LLM is more likely to be misled by human answers and even ignore reasoning processes written by humans. Whether the answer is correct or not, LLM tends to identify the answer as incorrect. As shown in Figure 7, the model misidentifies the answer of the human as "$8000". This may be because humans sometimes omit mathematical conversions in their answers, or because the problem-solving strategy may differ from that of the LLM. This can confuse the LLM. For instance, in Figure 8, the model fails to comprehend the equations without natural language explanations, erroneously identifies the human's problem-solving approach as incorrect, and proceeds to solve it using its own method. Even when provided with a rationale, the model usually fails to make an accurate identification, often leading to erroneous outcomes.

## F Type of Generalization

Table 9 shows the type of generalization we are investigating, based on the taxonomy defined by Hupkes et al. (2022).

'Problem': 'how much space , in cubic units , is left vacant when maximum number of 3 x 3 x 3 cubes are fitted in a rectangular box measuring 6 x 9 x 11 ?'

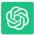 To solve this problem, we first need to determine how many 3 x 3 x 3 cubes can fit inside the given rectangular box.

The length of the box is 11 units, which is equal to 3 + 3 + 3 + 2 (two cubes fit partially). The width of the box is 9 units, which is equal to 3 + 3 + 3. And the height of the box is 6 units, which is equal to 3 + 3.

Therefore, we can fit 4 cubes in the length, 3 cubes in the width, and 2 cubes in the height, giving a total of 4 x 3 x 2 = 24 cubes.

The total volume of the cubes is 24 x (3 x 3 x 3) = 648 cubic units.

The volume of the rectangular box is 6 x 9 x 11 = 594 cubic units.

Therefore, the vacant space left in the box is:

594 - 648 = -54 cubic units

This means that there is no vacant space left in the box, and the cubes fill it completely with no room to spare.

Figure 3: Misconception in a three-dimensional question

'Problem': 'two numbers are in the ratio of 1 : 2 . if 5 be added to both , their ratio changes to 3 : 5 . the greater number is'

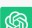 Let's assume the two numbers in the ratio of 1:2 are x and 2x, respectively.

According to the problem, when 5 is added to both numbers, their ratio becomes 3:5. So we can set up the following equation:

(x + 5) / (2x + 5) = 3/5

To solve for x, we can cross-multiply and simplify:

5(x + 5) = 3(2x + 5)
5x + 25 = 6x + 15
x = 10

Therefore, the two numbers in the ratio of 1:2 are 10 and 20. To find the greater number, we can simply add 5 to each and compare:

10 + 5 = 15
20 + 5 = 25

So the greater number is 25.

Figure 4: The GPT-3.5 model misinterpreting a mathematical question

'Problem': 'a 75 - liter solution of cool - drink is made from 8 % jasmine water . if 3 liters of jasmine and 12 liters of water were added to the solution , what percent of the solution is jasmine ?'

Student:
"To solve the problem, we can use the concept of the amount of jasmine water in the original solution and the amount of jasmine water in the final solution.

Initially, the solution contains 8% jasmine water, which means that in 75 liters of solution, there are 0.08 x 75 = 6 liters of jasmine water.

When 3 liters of jasmine water are added to the solution, the total amount of jasmine water becomes 6 + 3 = 9 liters.

Similarly, when 12 liters of water are added to the solution, the total volume of the solution becomes 75 + 3 + 12 = 90 liters."

You are a math teacher,
verify whether this student is wrong ,
if he is wrong,
correct it and give him explanation

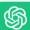 The student's initial calculations are correct. However, their final conclusion is missing.

To determine the percentage of jasmine water in the final solution, we need to calculate the amount of jasmine water as a percentage of the total solution.

The total amount of jasmine water in the final solution is 9 liters, which is 9/90 = 0.1 or 10% of the final solution.

Therefore, the answer is that the final solution contains 10% jasmine water.

So, the student needs to add a final step to their solution to calculate the percentage of jasmine water in the final solution.

Corrected solution:

Initially, the solution contains 8% jasmine water, which means that in 75 liters of solution, there are 0.08 x 75 = 6 liters of jasmine water.

When 3 liters of jasmine water are added to the solution, the total amount of jasmine water becomes 6 + 3 = 9 liters.

Similarly, when 12 liters of water are added to the solution, the total volume of the solution becomes 75 + 3 + 12 = 90 liters.

Therefore, the final solution contains 10% jasmine water (9 liters of jasmine water out of 90 liters of the total solution).

Figure 5: Completing the problem-solving process in the teacher–student framework

'Problem': 'the ratio between the number of sheep and the number of horses at the stewar farm is 3 to 7 . if each of horse is fed 230 ounces of horse food per day and the farm needs a total 12880 ounces of horse food per day . what is number sheep in the form ? ?'

Student:
"Let the number of sheep be 3x and the number of horses be 7x. Then the total number of animals on the farm is 3x + 7x = 10x.

Since each horse is fed 230 ounces of horse food per day, the total amount of horse food needed for 7x horses is 7x * 230 = 1610x ounces per day.

We are given that the total amount of horse food needed for the entire farm is 12880 ounces per day. So we can set up the following equation:

1610x = 12880

Solving for x, we get:

x = 8

Therefore, the number of sheep on the farm is 3x = 3*8 = 24.

So, the answer is 24."

You are a math teacher,
verify whether this student is wrong ,
if he is wrong,
correct it and give him explanation

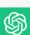

The student's work is correct up to the point of determining the value of x as 8. However, their final answer is incorrect.

If there are 7x horses on the farm, and we know that x is 8, then there are 7x = 7*8 = 56 horses on the farm.

The ratio of sheep to horses is 3:7, which means that for every 3 sheep, there are 7 horses. Since there are 56 horses, we can set up the following equation to find the number of sheep:

3/7 = x/56

Cross-multiplying, we get:

7x = 3*56

Simplifying, we get:

7x = 168

Dividing both sides by 7, we get:

x = 24

Therefore, the number of sheep on the farm is 3x = 3*24 = 72.

So, the correct answer is 72 sheep.

Figure 6: Providing the wrong reasoning process in the teacher–student framework

'Problem': 'a person lent a certain sum of money at 5 % per annum at simple interest and in 8 years the interest amounted to $ 480 less than the sum lent . what was the sum lent ?'

Student:
"Assume the amount of money lent is represented by X
(X*0.05)8+480=X
0.4X+480=X
=>0.6X=480
X=800"

You are a math teacher,
verify whether this student is wrong ,
if he is wrong,
correct it and give him explanation

The student's answer is incorrect. Let's go through the problem and solve it step by step.

Let's assume the amount of money lent is represented by 'X.'

According to the problem, the interest earned in 8 years is $480 less than the sum lent. So, the interest earned is X - $480.

Now, we can use the formula for simple interest:

Simple Interest = Principal * Rate * Time

In this case, the principal is X, the rate is 5% (0.05 as a decimal), and the time is 8 years.

Therefore, we can write the equation as:

(X * 0.05 * 8) = (X - $480)

Simplifying the equation, we have:

0.4X = X - $480

To solve for X, we need to isolate it on one side of the equation. Let's subtract X from both sides:

0.4X - X = - $480

-0.6X = - $480

Now, we can divide both sides by -0.6 to find the value of X:

X = (- $480) / (-0.6)

Simplifying the right side of the equation, we get:

X = $800

So, the correct answer is that the sum lent was $800, not $8000 as the student suggested.

Figure 7: Misidentification of the human's answer

'Problem': 'a person lent a certain sum of money at 5 % per annum at simple interest and in 8 years the interest amounted to $ 480 less than the sum lent . what was the sum lent ?'

Student:
"0.05X*8=X-480
0.6X=480
X=800"

You are a math teacher,
verify whether this student is wrong ,
if he is wrong,
correct it and give him explanation

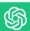

The student made a mistake in setting up the equation. Let's go through the problem and correct the error.

Let's assume the sum of money lent is 'X' dollars.

According to the problem, the interest earned after 8 years is $480 less than the sum lent. So, the interest earned is (X - $480).

The formula for simple interest is: Interest = Principal * Rate * Time

In this case, the rate is given as 5% per annum, which can be written as 0.05. The time is 8 years.

Therefore, the interest earned can be calculated as 0.05 * X * 8 = 0.4X.

Now, we can set up the equation based on the given information:

0.4X = X - $480

To solve for X, let's bring all the terms involving X to one side of the equation:

0.4X - X = -$480

Simplifying, we get:

-0.6X = -$480

Now, we can multiply both sides of the equation by -1 to get rid of the negative sign:

0.6X = $480

Finally, dividing both sides of the equation by 0.6 gives us:

X = $800

Therefore, the correct answer is that the sum lent was $800.

Explanation:
The mistake the student made was in the initial equation setup. They subtracted $480 from the interest side of the equation, which led to an incorrect equation. It's important to set up the equation accurately to solve the problem correctly.

Figure 8: Misinterpretation of the human's answer

| Motivation | | | |
|---|---|---|---|
| *Practical* | *Cognitive* | *Intrinsic* | *Fairness* |
| O | | | |
| **Generalisation type** | | | |
| *Compositional* | *Structural* | *Cross Task* | *Cross Language* *Cross Domain* | *Robustness* |
| | | | O | O |
| **Shift type** | | | |
| *Covariate* | *Label* | *Full* | *Assumed* |
| O | | | |
| **Shift locus** | | | |
| *Train–test* | *Finetune train–test* | *Pretrain–train* | *Pretrain–test* |
| | | | O |

Table 9: GenBench evaluation card