# OpenReview forum: "Three Questions Concerning the Use of Large Language Models to Facilitate Mathematics Learning"
_EMNLP/2023/Conference — EMNLP 2023 Findings_

### Official Review · Reviewer_pKga · 2023-08-04

**Soundness:** 3

**Excitement:**

3: Ambivalent: It has merits (e.g., it reports state-of-the-art results, the idea is nice), but there are key weaknesses (e.g., it describes incremental work), and it can significantly benefit from another round of revision. However, I won't object to accepting it if my co-reviewers champion it.

**Paper Topic And Main Contributions:**

This study introduced a research agenda on the utilization of Large Language Models (LLMs) in mathematical education. It raised a set of research questions, which were qualitatively and quantitatively derived. First, this study argued for the utility of LLMs in mathematical education through reviewing the previous research on automatic evaluation methods. Then, it analyzed the LLM's errors in answers for mathematical problem-solving questions. It also evaluated the error-correction ability of LLMs. These findings indicated the potential for the research agenda regarding educational utilization of LLMs.

**Reasons To Accept:**

This study presents a research agenda for the effective use of LLMs in mathematical education. It also explores the advantages and disadvantages of utilizing LLMs in this context. This study has also explored the LLM's capability to correct students' problem-solving processes.

**Reasons To Reject:**

This study carried out an experiment using an LLM both as a student for answering questions and as a teacher for correcting students' answers and providing feedback on incorrect answers. This approach is understandable as a starting point. The author(s) mentioned room for improvement in LLM performance.

**Reproducibility:**

3: Could reproduce the results with some difficulty. The settings of parameters are underspecified or subjectively determined; the training/evaluation data are not widely available.

**Reviewer Confidence:**

3: Pretty sure, but there's a chance I missed something. Although I have a good feel for this area in general, I did not carefully check the paper's details, e.g., the math, experimental design, or novelty.

---

> ### Author Rebuttal · Authors · 2023-08-23
>
> Q3-1: it would have been beneficial to present the validity of the LLM-generated answers as alternatives to students' answers.
>
> A3-1: I do agree with your perspective. We have indeed explored the LLM's capability to correct students' problem-solving processes, as detailed in Appendix E. Our choice to use LLM-generated answers was primarily driven by the need to obtain a large number of problem-solving processes to facilitate an analysis at a certain scale. Moreover, we were curious if the machine could comprehend its answers. If the LLM struggles to understand its responses (with detailed processes and natural language explanations), then its potential to assist teachers in corrections becomes questionable. The experimental results in Table 4 show that even when provided with the correct answer for a question, the LLM is unable to correct it effectively.

---

### Official Review · Reviewer_RNa9 · 2023-08-05

**Soundness:** 3

**Excitement:**

2: Mediocre: This paper makes marginal contributions (vs non-contemporaneous work), so I would rather not see it in the conference.

**Paper Topic And Main Contributions:**

Aiming for understand "how can LLM be utilized to assist students in learning mathematical problems skills", this paper investigated two sub problems:
1) How much LLM can solve the math problems with computational process
2) How much LLM can check the correctness of a given answer and generate feedback.

This paper utilized GPT-3.5 as the LLM, used MathQA dataset to run experiments.
1. They reported the correctness of how much ChatGPT can answer the questions in MathQA;
2. They used a math expert to categorize the errors made by LLM;
3. They used a teacher-student prompt to check if LLM can identify the correctness of a given answer then provided categorized error type for the errors.

Contributions:
1. A proposal to breaking LLM for Math Education problems into two sub problems
2. Ran experiment to show where GPT3.5 is in solving these problems, and what types of errors GPT3.5 is making
3. The teacher-student prompting to get LLM to check if a given answer is correct is a bit interesting



**Reasons To Accept:**

1) Contribution above is interesting. It made an abstract question more concrete.
2) Used a dataset and experiments to quantify the problems and issues.

**Reasons To Reject:**

1. The author considered this as a proposal of "how to use LLM for math problem solving". However, I only see the author shared where the current status is with GPT3.5, I am not seen much of about the "how" part being discussed.
2. The experiments' findings aren't new. It seems to be known that LLM isn't for solving math problems. Categorizing the errors provided some new data points. How were these categories developed and why they are such categories? And what we should do these categories of errors?
3. The categorization of errors were done by 1 person. How do we ensure the subjectivity not impacting the categorization or accuracy?
Based on above, I am not if there are enough contributions.

**Reproducibility:**

5: Could easily reproduce the results.

**Reviewer Confidence:**

4: Quite sure. I tried to check the important points carefully. It's unlikely, though conceivable, that I missed something that should affect my ratings.

---

> ### Author Rebuttal · Authors · 2023-08-23
>
> Q2-1: The author considered this as a proposal of "how to use LLM for math problem solving". However, I only see the author shared where the current status is with GPT3.5, I am not seen much of about the "how" part being discussed.
>
> A2-1: I understand why you have this doubt. The primary focus of our paper is to discuss the potential issues of directly applying LLMs in an educational context like GPT-3.5. We use the math problem-solving task as a means to study the types of errors LLMs might commit. Given that LLMs are among the most accessible tools for the general public and many educational institutions are exploring the challenges and benefits of teachers and students using LLMs, our intention with this paper is to explore these related topics through some experiments. We will add this clarification in our final version.
>
> Q2-2: It seems to be known that LLM isn't for solving math problems.
>
> A2-2: Given the remarkable explanatory capabilities of LLMs, we believe they have notable potential to help students and enhance their learning processes. Thus, our work starts by analyzing the LLMs' capabilities in solving mathematical problems and then examines the weaknesses in LLM's correction and explanation.
>
> Q2-3: How were these categories developed and why they are such categories? And what we should do these categories of errors?
>
> A2-3: The categories of errors were mainly defined by an expert who majored in mathematics and graduated from a university of education. We invited the expertise of this specialist to categorize the errors made by the model. While we know the limitations of LLMs in various tasks, our primary goal was to closely examine the nature of errors that LLMs make in problem-solving and correction. We hope the insight from our experiments will facilitate future research on the deployment of LLMs within educational domains. We will add the explanation in our final version. Thanks for your comment.
>
> Q2-4: The categorization of errors were done by 1 person. How do we ensure the subjectivity not impacting the categorization or accuracy?
>
> A2-4: We'd like to note that asking an expert with both mathematical and educational expertise willing to assist in this endeavor was a challenge, and we hope you take this into consideration. But I completely understand that this is a reasonable concern.

---

### Official Review · Reviewer_PZHc · 2023-08-05

**Typos Grammar Style And Presentation Improvements:** N/A
**Soundness:** 4

**Excitement:**

4: Strong: This paper deepens the understanding of some phenomenon or lowers the barriers to an existing research direction.

**Missing References:**

N/A

**Paper Topic And Main Contributions:**

# Summary

In this work, the authors explore answers to three simple questions -- Use of LLMs in Math Learning,the mathematical and problem-solving abilities of LLMs and the use of LLMs to identify errors in student solutions to math problems.  Even as they may be simple questions to ask, the paper does a fair job at analyzing the answers to these questions. For the first question, the paper provides literature review evidence on the use of LLMs as assessment tools. For the second question, the paper use a modified version of the MathQA dataset to test GPT models with automated and human evaluation to identify inconsistencies.

**Questions For The Authors:**

1. What would be another question that could asked in a somewhat related context, however extending the current discussion into a deeper aspect in terms of reasoning of LLMs? could this extended discussion be also included in the current submission?

**Reasons To Accept:**

## Reasons

There are a few reasons to accept the work, surrounding the thoughtful nature of the analysis. The authors take time to breakup the analysis into three separate steps to understand the internals of LLMs. Specifically,

1. With a deeper understanding and with an answer to RQ2, the authors able to categorize the responses that LLMs generate as explanations when solving math problems. This may better help in understanding the development of cognitive tutors with LLMs

2. With a general understanding on the limitations and capacities of LLMs, it is a great way to scope the expectations that people seem to be developing surrounding the hype around LLMs.

**Reasons To Reject:**

## Reasons

The main weakness of this paper is the answer of RQ3. With discussions on the teacher-student framework for quantitative evaluation, to compare with correcting model generated answers, there is a certain possibility to understand if LLMs can act as a tools to help students correct errors. However, the discussion on reasoning task may be a little far--fetched with just this analysis. It is important to develop a cognitive model for the process of reasoning and the roles it could play in the process of correcting student errors. However, this may be a minor aspect to discuss here.

quick reference Lovett, Marsha C. "Cognitive task analysis in service of intelligent tutoring system design: A case study in statistics." International Conference on Intelligent Tutoring Systems. Berlin, Heidelberg: Springer Berlin Heidelberg, 1998.

**Reproducibility:**

3: Could reproduce the results with some difficulty. The settings of parameters are underspecified or subjectively determined; the training/evaluation data are not widely available.

**Reviewer Confidence:**

3: Pretty sure, but there's a chance I missed something. Although I have a good feel for this area in general, I did not carefully check the paper's details, e.g., the math, experimental design, or novelty.

---

> ### Author Rebuttal · Authors · 2023-08-23
>
> Q1-1: The main weakness of this paper is the answer of RQ3. The discussion on reasoning task may be a little far--fetched with just this analysis.
>
>
> A1-1: Thanks for your comment. We acknowledge the importance of developing a cognitive model for reasoning. At the current stage, we primarily focus on the capabilities and challenges of directly using LLMs for correcting students’ answers. But the weakness you mentioned is important. We'll address this limitation in our final version.
>
>
>
> Q1-2: What would be another question that could asked in a somewhat related context, however extending the current discussion into a deeper aspect in terms of reasoning of LLMs? could this extended discussion be also included in the current submission?
>
>
> A1-2: We believe that a deeper discussion on the reasoning of LLMs is feasible based on the results and observations of our experiments. We will extend this idea to conduct comprehensive research at the next stage. Thank you for your suggestion.

---

### Meta-Review · Area_Chair_Ggwd · 2023-09-10

**Recommendation:** 4

**Metareview:**

The main conclusions of the reviews and the post-rebuttal discussions:
- 3/ 3 reviewers consider the paper sound (scores 4, 3, 3)
- 2/ 3 reviewers find the paper exciting (scores 4, 2, 3)

Front reading the rebuttal and seeing the scores above, I find that the reviewers consider strong points for soundness the following:
- In-depth exploration of 3 RQs, on the math and problem-solving abilities of LLMs and the use of LLMs to identify errors in student solutions to math problems
- especially RQ1 and RQ2 seem to contain valuable analysis, in particular for developing better tutors with LLMs
Concerns regarding the use of just 1 annotator were clarified in the rebuttal as it is challenging to find an expert in both math and education.

---

### Decision · Program_Chairs · 2023-10-07

**Decision:**

Accept-Findings

**Comment:**

The main conclusions of the reviews and the post-rebuttal discussions:
- 3/ 3 reviewers consider the paper sound (scores 4, 3, 3)
- 2/ 3 reviewers find the paper exciting (scores 4, 2, 3)

Front reading the rebuttal and seeing the scores above, I find that the reviewers consider strong points for soundness the following:
- In-depth exploration of 3 RQs, on the math and problem-solving abilities of LLMs and the use of LLMs to identify errors in student solutions to math problems
- especially RQ1 and RQ2 seem to contain valuable analysis, in particular for developing better tutors with LLMs
Concerns regarding the use of just 1 annotator were clarified in the rebuttal as it is challenging to find an expert in both math and education.